# Adaptive Pruning of Neural Language Models for Mobile Devices

## Abstract

Neural language models (NLMs) exist in an accuracy–efficiency tradeoff space where better perplexity typically comes at the cost of greater computation complexity. In a software keyboard application on mobile devices, this translates into higher power consumption and shorter battery life. This paper represents the first attempt, to our knowledge, in exploring accuracy–efficiency tradeoffs for NLMs. Building on quasi-recurrent neural networks (QRNNs), we apply pruning techniques to provide a "knob" to select different operating points. In addition, we propose a simple technique to recover some perplexity using a negligible amount of memory. Our empirical evaluations consider both perplexity as well as energy consumption on a Raspberry Pi, where we demonstrate which methods provide the best perplexity–power consumption operating point. At one operating point, one of the techniques is able to provide energy savings of 40% over the state of the art with only a 17% relative increase in perplexity.

## 1 Introduction

An emerging application of neural language models (NLMs) is smart software keyboards on such mobile devices as smartphones and tablets that provide next-word prediction, allowing users to input entire words with a single tap. For example, the app SwiftKey[1] advertises the use of neural networks for predictions;[2] according to Google Play Store, it has more than 100 million downloads, demonstrating its popularity.

Based on standard metrics such as perplexity, neural techniques represent an advance in the state of the art in language modeling (Merity et al., 2018b). Better models, however, come at a cost in computational complexity, which translates to higher power consumption (Tang & Lin, 2018). In the context of mobile devices, energy efficiency is, of course, an important optimization objective. A casual web search, for example, reveals numerous complaints from users of the above apps about battery drain, indicating that this is not a hypothetical concern.

In reality, neural language models exist in a accuracy–efficiency tradeoff space. Although this fact has been recognized for applications such as image recognition (Canziani et al., 2016) and keyword spotting (Tang et al., 2018), to our knowledge no one in the NLP community has explored these tradeoffs. All previous papers on NLMs simply report single-point perplexity figures. In contrast, the high-level goal of our work is to understand the tradeoffs between neural modeling accuracy and real-world efficiency constraints: in addition to perplexity, NLMs should be evaluated in terms of FLOPs,[3] milliJoule per query (mJ/q), and inference latency. We conduct exactly such experiments, using the Raspberry Pi (which shares the same architecture as most mobile devices today) as a more convenient hardware platform.

Ideally, NLMs should provide a "knob" that allows developers to tune accuracy–efficiency tradeoffs. In this paper, we explore pruning approaches that take a pre-trained quasi-recurrent neural network (QRNN; Bradbury et al., 2017), representing the state of the art in NLM today, and provides exactly such a knob. Furthermore, our techniques allow these tradeoffs to be tuned at *inference time*, which

---

[1] http://www.swiftkey.com/

[2] https://blog.swiftkey.com/swiftkey-debuts/

[3] Convention from literature defines number of FLOPs as the total number of additions and multiplications.

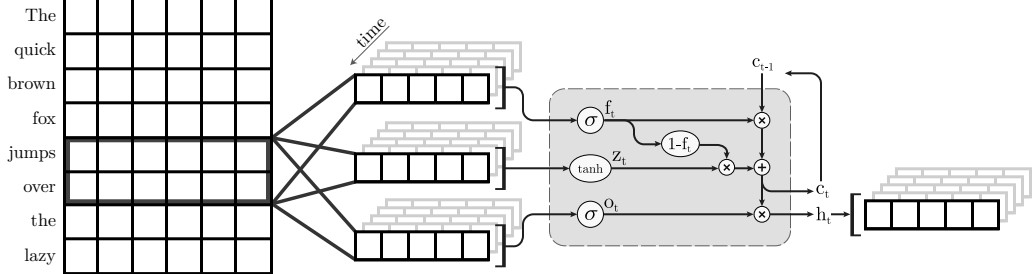

Figure 1: An illustration of the first QRNN layer for language modeling. In this visualization, a QRNN layer with a window size of two convolves and pools using embeddings from the input. Note the absence of recurrent weights.

allows a mobile device to adaptively control its behavior, e.g., favor efficiency at the cost of accuracy when the battery is low.

Thus, this paper makes the following contributions: First, to our knowledge, we are the first to comprehensively explore accuracy–efficiency tradeoffs for NLMs with experimental evaluation of energy consumption on a Raspberry Pi. Second, we evaluate a number of inference-time pruning techniques that takes any pre-trained QRNN and provides a tunable accuracy–efficiency "knob".

## 2 BACKGROUND AND RELATED WORK

### 2.1 QUASI-RECURRENT NEURAL NETWORKS

Quasi-recurrent neural networks (QRNNs; Bradbury et al., 2017) achieve highly competitive perplexity on word-level language modeling datasets, including state-of-the-art perplexity on WikiText-103 (Merity et al., 2018b). Although applying such techniques as dynamic evaluation (Krause et al., 2017), Hebbian softmax (Rae et al., 2018), and mixture of softmaxes (Yang et al., 2017) can produce lower perplexity, our focus is on the recurrent architecture. Thus, we explore the task of pruning QRNNs without using any other extensions.

Each word is encoded as a one-hot vector and then fed into a linear layer, which produces lower-dimensional word embeddings for the QRNN layers. A single QRNN layer consists of two distinct components—convolution and recurrent pooling—that alternate to imitate an LSTM (Hochreiter & Schmidhuber, 1997). Given a stacked sequence of inputs $\mathbf{X} = \boldsymbol{x}_1 \oplus \cdots \oplus \boldsymbol{x}_n \in \mathbb{R}^{k \times n}$ (e.g., word embeddings in language modeling), the one-dimensional convolution layer is defined as

$$\mathbf{Z} = \tanh(\mathbf{W}_z \cdot \mathbf{X})$$
$$\mathbf{F} = \sigma(\mathbf{W}_f \cdot \mathbf{X})$$
$$\mathbf{O} = \sigma(\mathbf{W}_o \cdot \mathbf{X})$$

where $\mathbf{W}_z, \mathbf{W}_f, \mathbf{W}_o$ are the weights associated with the input, forget, and output gates, respectively, $\cdot$ represents a masked convolution along time, and $\sigma$ denotes the sigmoid function. For $\mathbf{W}_{\{z,f,o\}} \in \mathbb{R}^{m \times (k \times r)}$, $m$ is the number of output channels, $k$ is the number of input channels, and $r$ the window size across time. Without loss of generality, we henceforth represent $\mathbf{W}_{\{z,f,o\}}$ as two-dimensional matrices $\in \mathbb{R}^{m \times s}$, where $s = k \times r$. The outputs are fed into a recurrent pooling layer:

$$\mathbf{c}_t = \mathbf{f}_t \odot \mathbf{c}_{t-1} + (1 - \mathbf{f}_t) \odot \mathbf{z}_t$$
$$\mathbf{h}_t = \mathbf{o}_t \odot \mathbf{c}_t$$

where $\odot$ denotes element-wise product. Altogether, these two layers define a single QRNN layer (Bradbury et al., 2017; see Figure 1). Multiple layers can be stacked for greater expressiveness, where the output $\mathbf{h}_{1:n}$ of the previous layer is the input $\mathbf{X}$ to the current layer.

We tie the weights between the input and output layers, as used by Merity et al. (2018a) and proposed by Inan et al. (2017). In addition to improving perplexity, weight tying reduces the number of parameters and hence the memory footprint, which is beneficial to our task.

## 2.2 PRUNING

Weight pruning is an effective strategy for reducing the computational footprint of a model. An influential pioneering work, LeCun et al. (1990) proposes to discard weights using a error-approximation approach based on Hessian diagonals. More recent work suggests pruning weights with small magnitudes (Han et al., 2016), with quantization and Huffman coding as additional steps. However, these approaches introduce irregular sparsity to the weights, and they assume that re-training the weights is feasible.

In this work, we take a different approach and focus on techniques that eliminate entire filters. This is because modern implementations of feedforward evaluation (e.g., im2col and particularly NEON instruction on ARM processors) take advantage of dense matrix multiplications. Pruning individual weights without changing the dimensions of the weight matrices has minimal effect on power consumption—this is confirmed by our initial exploratory studies on the Raspberry Pi. Hence, we only examine pruning techniques that discard entire filters of the convolutional layers:

**Random pruning.** A simple baseline (Mittal et al., 2018) is random filter pruning, where $n\%$ of the filters are randomly pruned, layer-by-layer. Interestingly, Mittal et al. (2018) find that random pruning is competitive with more advanced methods.

**Filter norm.** Li et al. (2017) propose ranking filters by their $L_1$-norms, and then dropping off $n\%$ of the smallest filters on a layer-by-layer basis. Mittal et al. (2018) have previously found that $L_1$-norm filter pruning (Li et al., 2017) outperforms a multitude of competing approaches.

**Mean activation norm.** Among other approaches, Molchanov et al. (2016) suggest pruning filters whose mean activations are small. This approach is especially effective on ReLU, which both creates sparse activations and forces them to be non-negative.

$L_0$ **regularization.** Louizos et al. (2018) apply $L_0$ regularization to neural networks in order to learn sparse, efficient structures. Formally, define an objective

$$\mathcal{R}(\boldsymbol{\theta}) = \mathcal{L}(\boldsymbol{\theta}) + \lambda\|\boldsymbol{\theta}\|_0$$
$$\boldsymbol{\theta}^* = \arg\min_{\boldsymbol{\theta}} \mathcal{R}(\boldsymbol{\theta})$$

where $\mathcal{L}$ is the original loss function and $\boldsymbol{\theta}$ the weights. The dependence on the hypothesis and training examples has been omitted for brevity. The optimal solution entails a non-differentiable objective and iteration over all $2^{|\boldsymbol{\theta}|}$ possibilities to choose the best $\boldsymbol{\theta}^*$; hence, Louizos et al. (2018) propose the following relaxation of the objective:

$$\hat{\mathcal{R}}(\boldsymbol{\theta}, \boldsymbol{\phi}) = \mathbb{E}_{\mathbf{z}\sim p(\mathbf{z}|\boldsymbol{\phi})}\left[\mathcal{L}(\boldsymbol{\theta}\odot\mathbf{z})\right] + \lambda\sum_{i=1}^{|\boldsymbol{\theta}|}\left(1 - Q(\mathbf{z}_i \leq 0; \boldsymbol{\phi}_i)\right)$$
$$\boldsymbol{\theta}^*, \boldsymbol{\phi}^* = \arg\min_{\boldsymbol{\theta},\boldsymbol{\phi}} \hat{\mathcal{R}}(\boldsymbol{\theta}, \boldsymbol{\phi})$$

where $\mathbf{z} \sim p(\mathbf{z}|\boldsymbol{\phi})$ is a binary discrete random mask parameterized by $\boldsymbol{\phi}$, and $Q$ is the CDF. Intuitively, for some choice of $\boldsymbol{\phi}$, the number of active parameters (on average) is penalized. Inspired by the Concrete distribution (Maddison et al., 2016), Louizos et al. (2018) propose the hard concrete distribution for $\mathbf{z}$, further relaxing the discrete random mask into a continuous one:

$$\mathbf{s} = \sigma\left((\log\mathbf{u} - \log(1 - \mathbf{u}) + \log\boldsymbol{\alpha})/\beta\right)$$
$$\mathbf{z} = \min(\mathbf{1}, \max(\mathbf{0}, (\zeta - \gamma)\mathbf{s} + \gamma))$$

where $\mathbf{u} \in \mathbb{R}^{|\boldsymbol{\theta}|}$ is a continuous random vector such that $\mathbf{u}_i \sim \text{Uniform}[0, 1]$, $\boldsymbol{\phi} = \log\boldsymbol{\alpha}$ are the mask parameters, and $\gamma = -0.1, \zeta = 1.1, \beta = 2/3$ are scaling hyperparameters. Note that $\beta$ can also be included as part of the mask parameters $\boldsymbol{\phi}$; we follow Louizos et al. (2018) and fix $\beta = 2/3$. Louizos et al. (2018) then apply the reparameterization trick (Kingma & Welling, 2014; Rezende et al., 2014) and make a Monte Carlo approximation to the objective:

$$\hat{\mathcal{R}}(\boldsymbol{\theta}, \boldsymbol{\phi}) = \frac{1}{N}\sum_{i=1}^{N}\left(\mathcal{L}(\boldsymbol{\theta}\odot\mathbf{z}^{(i)})\right) + \lambda\sum_{i=1}^{|\boldsymbol{\theta}|}\left(1 - Q(\mathbf{z}_i \leq 0; \boldsymbol{\phi}_i)\right)$$

A closed form expression is derived for the penalty, $(1 - Q(\mathbf{z}_i \leq 0; \boldsymbol{\phi}_i)) = \sigma(\log \boldsymbol{\alpha}_i - \beta \log \frac{-\gamma}{\zeta})$. At test time, the following estimator is used:

$$\mathbf{z} = \min(\mathbf{1}, \max(\mathbf{0}, \sigma(\log \boldsymbol{\alpha})(\zeta - \gamma) + \gamma)$$

## 3  INFERENCE-TIME PRUNING

In this section, we explain how the various techniques in Section 2.2 can be adapted to QRNNs. For the following methods, we assume that a pre-trained model is provided. We denote the weights at QRNN layer $l$ as $\mathbf{W}^{(l)}$. In all methods, we tie the indices across $\mathbf{W}_z, \mathbf{W}_f, \mathbf{W}_o$. For example, if filter $i$ is selected for pruning at layer $l$, then $\mathbf{W}_{\{z,f,o\}}^{(l)} := \mathbf{W}_{\{z,f,o\}}^{(l)}[-i, :]$, where $-i$ denotes exclusion of index $i$. This allows the removal of the column $[:, -i]$ in the next layer as well.

**Random pruning.** We apply random pruning to $\mathbf{W}_z$, $\mathbf{W}_f$, and $\mathbf{W}_o$. That is, we randomly prune filters associated with the same indices across the three weights.

**Filter norm.** We apply filter norm pruning (Li et al., 2017), with the filter norms of $\mathbf{W}_z$ acting as the criteria. We find $\mathbf{W}_z$ most helpful, since small filter norms should result in small hidden outputs, which is not necessarily the case for $\mathbf{W}_f$ and $\mathbf{W}_o$.

**Mean activation norm.** The hidden output $\mathbf{H} = \mathbf{h}_1 \oplus \cdots \oplus \mathbf{h}_n$ is a natural candidate for collecting mean activation statistics. Intuitively, if $\|\mathbf{H}_{:,i}\|_1$ is small on average, then the $i^{th}$ filters for $\mathbf{W}_z, \mathbf{W}_f, \mathbf{W}_o$ are less important. Statistics are collected using a single pass of the entire training set. For inference-time pruning, we store the collected statistics.

$L_0$ **regularization.** Since we are given a pre-trained model and are prohibited from altering the weights, we learn the mask parameters only: $\boldsymbol{\phi}^* = \arg\min_{\boldsymbol{\phi}} \hat{\mathcal{R}}(\boldsymbol{\theta}, \boldsymbol{\phi})$. We also enforce the sparsity on entire rows of $\mathbf{W}_z$, which corresponds to "group sparsity" in Louizos et al. (2018). Specifically, we formulate the regularization on a feature map level instead, with $\mathbf{Z}$ as the target:

$$\mathbf{Z}^{(l)} := \left( \text{diag}(\mathbf{z}^{(l)}) \mathbf{W}_z^{(l)} \right) \cdot \mathbf{X} = \mathbf{Z}^{(l)} \odot \mathbf{z}^{(l)}$$

$\mathbf{Z}$ is chosen for the property that the $i^{\text{th}}$ feature map for $\mathbf{h}$ is zero if $\mathbf{Z}_i$ is zero for $\mathbf{c}_0 = \mathbf{0}$.

This approach entails training and storing extra mask parameters for each operating point. However, we find this to be a non-issue for our task, since there are few operating points—three or four at most, out of which we use two for $L_0$ regularization—so the extra storage is negligible.

### 3.1  WITH SINGLE-RANK UPDATE

At specific operating points (e.g., 40% and 80% FLOPs), pre-trained weight updates can be stored and applied at inference-time to recover some perplexity. Suppose $\mathbf{W} \in \mathbb{R}^{m \times n}$ is a weight matrix in a neural network, and $\mathbf{W}^* \in \mathbb{R}^{m \times n}$ is some known set of weights that results in a lower loss. Clearly, $\Delta\mathbf{W} := \mathbf{W}^* - \mathbf{W}$ can be stored and added at inference-time to obtain a better neural network. However, it is obvious that this scheme is wasteful, since $\mathbf{W}^*$ could have directly substituted $\mathbf{W}$ in the first place.

Sacrificing a negligible amount of storage to recover some perplexity, we propose learning a single-rank weight matrix update

$$\Delta\mathbf{W} := \mathbf{u}\mathbf{v}^{\mathsf{T}}, \mathbf{u} \in \mathbb{R}^m, \mathbf{v} \in \mathbb{R}^n$$

to each weight in the convolution layers. Specifically, the process is as follows, beginning with a pre-trained model:

1. Prune a pre-determined set of filters for some operating point (e.g., 40% FLOPs).

2. Initialize the weight updates $\Delta\mathbf{W}_l = \mathbf{u}^{(l)}\mathbf{v}^{(l)\mathsf{T}}, \mathbf{u}_i^{(l)}, \mathbf{v}_i^{(l)} \sim p(\epsilon)$ for each convolution layer $l$, in our case Normal$(0, 0.1)$.

3. Fixing the existing weights $\mathbf{W}_l$ for each convolution layer, train a single-rank update such that $\mathbf{W}_l^* := \mathbf{W}_l + \Delta\mathbf{W}_l$, where $\mathbf{W}_l^*$ is used as the new weight.

4. Store $\Delta\mathbf{W}_l$ for use at inference time on the same operating point.

## 4 EXPERIMENTAL SETUP

We evaluate the aforementioned pruning techniques for word-level language modeling on Penn Treebank (PTB) (Marcus et al., 1993; as preprocessed by Mikolov et al., 2010) and WikiText-103 (WT-103) (Merity et al., 2017). We denote the models for PTB and WT-103 as `ptb-qrnn` and `wt103-qrnn`, respectively.

### 4.1 DATASETS AND TASKS

For each model, we report word-level perplexity and recall-at-three (R@3), defined as the percentage of top three token–logit outputs that contain the true next token—this was chosen due to most mobile keyboards providing three predictions. For example, if {"cat", "dog", "baby"} are the top three predicted tokens for, "I adopted a ___," with "dog" being the ground truth, then the prediction is correct, regardless of the rank of "dog".

**Penn Treebank.** Built from Wall Street Journal articles, Penn Treebank (PTB) is a small yet popular word-level dataset for language modeling. In the standard pre-processed version (Mikolov et al., 2010), the dataset contains roughly 887K, 70K, and 78K training, validation, and testing tokens, respectively. The number of unique tokens is capped at 10,000, yielding a relatively large 4.8% out-of-vocabulary (OOV) rate.

**WikiText-103.** Merity et al. (2017) introduce WikiText-2 and WikiText-103, datasets based on freely available Wikipedia articles. We use only WikiText-103, since WikiText-2 was designed to be similar to Penn Treebank. With 103 million training tokens, WikiText-103 is 103 times as large as PTB. WikiText-103 contains around 217K tokens for validation, and 245K for testing. The number of unique tokens is 267K, resulting in a 0.4% OOV rate, significantly lower than that of PTB.

### 4.2 HYPERPARAMETERS AND TRAINING

In all of the models, we chose the hyperparameters as suggested in Merity et al.'s codebase.[4] For `ptb-qrnn`, we used a four-layer QRNN with 1550 hidden units for each layer and a 400-dimensional embedding. For `wt103-qrnn`, we used a four-layer QRNN with 2500 hidden units and 400-dimensional embeddings, along with a tied adaptive softmax (Merity et al., 2018b). In both models, the first layer uses a window size of two, while the rest use a windows size of one.

Following Merity et al. (2018a), we also adopted the regularization techniques randomized back-propagation through time, embedding dropout, temporal activation regularization (TAR), activation regularization (AR), and variational dropout. We followed the same training process as well, with non-monotonically triggered ASGD (NT-ASGD) as the optimizer. We use the same hyperparameters as Merity et al. (2018a) and Merity et al. (2018b) for each model–dataset pair.

During the training of `wt103-qrnn`, we follow Merity et al. (2018b), using a tied adaptive softmax (Grave et al., 2017; Merity et al., 2018b) layer. At inference time, we use a regular softmax instead, since we require R@3.

**Pruning.** We selected a number of distinct operating points that represent discrete points in the accuracy–efficiency tradeoff space. Based on previous work (Tang et al., 2018), floating-point operations (FLOPs) is a good proxy of both energy usage and latency, and so we use FLOPs as a way of selecting our operating points. In $L_0$ regularization, the $\lambda$ decay strength was selected so that the resulting model corresponds to roughly the FLOPs targets: To achieve 80% and 60% FLOPs for the model on PTB, we used $\lambda = 5.5 \times 10^{-4}, 8.5 \times 10^{-4}$, respectively. To achieve about 70% FLOPs on WT-103, we chose $\lambda = 6 \times 10^{-4}$.

We trained the hard concrete mask parameters for roughly 5000 steps using Adam with a learning rate of $5 \times 10^{-3}$. Since the weight decay penalty is incompatible with the objective, we removed it while training the mask.

For mean activation pruning, which requires some training examples to collect statistics, we used the entire training set for `ptb-qrnn`. Since WikiText-103 is large, we used roughly 10% of the first training examples for collecting statistics on `wt103-qrnn`.

---

[4] `https://github.com/salesforce/awd-lstm-lm`

**Single-rank update (SRU).** For the PTB model, the single-rank update was trained for 10 epochs using NT-ASGD (Merity et al., 2018a) with a non-monotonic interval of three. For WikiText-103, the update was trained for 2000 steps using Adam with a learning rate of $5 \times 10^{-3}$. All other hyperparameters were the same as those used during the training stage.

### 4.3 INFRASTRUCTURE DETAILS

We trained all of our models on a commodity machine with a Titan V GPU, i7-4790k CPU, and 16 GB of RAM. We used PyTorch 0.4.0 (commit `1807bac`) for developing and running our models. We deployed our models on a Raspberry Pi (RPi) 3 Model B (ARM Cortex-A53) running Raspbian Stretch (4.9.41-v7+). Specifically, we copied the trained models over to the RPi, and ran them at the same operating points accordingly.

We plugged the RPi into a Watts Up Pro meter, a wattmeter that reports power usage at the rate of 1 Hz via a USB cable, which is connected back to the RPi. Evaluating on the test set, we collected power draw statistics on 350 next-word predictions, which were averaged to produce a millijoule per query (mJ/q) estimate. We obtained latency estimates in a similar manner by averaging the milliseconds per query (ms/q). Finally, we subtracted off the idle power usage of the RPi to obtain a better estimate of the actual power for each query.

Although our final application is NLMs running on mobile devices such as smartphones and tablets, there are many challenges to directly evaluating on such hardware, not to mention great variability between different smartphones. The Raspberry Pi is a standard, convenient stand-in since it uses exactly the same ARM processor architecture as nearly all mobile devices today, while being easy to develop on for practitioners. Evaluation on the RPi is widely adopted for research on efficient NNs today (Amato et al., 2017; Tang et al., 2018).

## 5 RESULTS AND DISCUSSION

In our results for PTB and WT-103, we compare to state-of-the-art results in the past. In general, we find that QRNNs are strong competitors to LSTM approaches, and achieve state-of-the-art perplexity on WikiText-103 (Merity et al., 2018b).

| # | Method | Model Quality | | | Footprint | | | w/SRU | |
|---|--------|------|------|------|---------|------|------|------|------|
| | | Val. | Test | R@3 | % FLOPs | ms/q | mJ/q | Test | R@3 |
| 1 | Skip LSTM | 60.9 | 58.3 | – | – | – | – | – | – |
| 2 | AWD-LSTM | 60.0 | 57.3 | – | – | 223 | 295 | – | – |
| 3 | Orig. | 59.0 | 56.8 | 44.7% | 100% | 224 | 296 | – | – |
| 4 | Orig. scratch | 61.5 | 59.3 | 44.0% | 80% | 182 | 238 | – | – |
| 5 | $L_0$ reg. | **63.0** | **60.7** | **43.6%** | 80% | 185 | 227 | **59.3** | **44.1%** |
| 6 | $L_0$ reg. | 69.2 | 66.8 | 42.1% | 60% | 142 | 183 | 64.0 | 42.7% |
| 7 | Random | 68.2 | 66.0 | 42.9% | 80% | 182 | 238 | 61.1 | 43.8% |
| 8 | Filter norm | 76.1 | 72.7 | 42.4% | 80% | 182 | 238 | 66.1 | 43.1% |
| 9 | Mean activation | 68.3 | 66.1 | 42.6% | 80% | 182 | 238 | 61.0 | 43.5% |

Table 1: Select pruning results on Penn Treebank using a 4-layer QRNN, along with past results drawn from the original papers. Skip LSTM refers to the four-layer skip LSTM from Melis et al. (2018), and AWD-LSTM is from Merity et al. (2018a). The four-layer QRNN (Merity et al., 2018b) is the same model that we use, but we achieve better perplexity following the same methodology. The best results of each category are bolded. "w/SRU" denotes the results after applying an SRU.

For PTB and WT-103, to demonstrate that the original model is well-parameterized, we train models with less parameters from scratch; see Tables 1 and 2, row 4—the smaller models achieve higher perplexity and lower R@3 than the originals do. We note that a 20-point increase in perplexity may only correspond to a few points decrease in R@3, showing that perplexity changes on a much different scale than accuracy does (see Table 1, rows 3 and 8). Furthermore, lower perplexity does not necessarily imply higher accuracy (see rows 6 and 8), confirming that perplexity alone cannot completely determine the recall. In Table 1, we chose 75 as the cutoff-point for perplexity—further

| # | Method | Model Quality | | | Footprint | | | w/SRU | |
|---|--------|------|------|------|---------|-------|------|------|------|
| | | Val. | Test | R@3 | % FLOPs | sec/q | J/q | Test | R@3 |
| 1 | Rae-LSTM | 36.0 | 36.4 | – | – | – | – | – | – |
| 2 | 4-layer QRNN | 32.0 | 33.0 | – | – | 1.24 | 1.48 | – | – |
| 3 | Orig. | 31.9 | 32.8 | 51.5% | 100% | 1.24 | 1.48 | – | – |
| 4 | Orig. scratch | 38.0 | 38.8 | 51.3% | 70% | 0.942 | 1.10 | – | – |
| 5 | $L_0$ reg. | **65.8** | **65.4** | **43.1%** | 69% | 0.912 | 1.06 | 56.9 | 44.7% |
| 6 | Mean activation | 89.8 | 92.9 | 38.9% | 70% | 0.942 | 1.10 | 55.7 | 46.0% |
| 7 | Filter norm | 85.9 | 88.2 | 41.7% | 70% | 0.942 | 1.10 | 59.2 | 45.4% |
| 8 | Random | 80.9 | 81.4 | 42.9% | 70% | 0.942 | 1.10 | **54.2** | **46.1%** |

Table 2: Select pruning results on WikiText-103 using a 4-layer QRNN, along with past results, drawn directly from the original papers. Note that Rae et al. (2018) primarily explore Hebbian softmax; Rae-LSTM refers to their LSTM model without any extensions. Bolded are the best results for each category.

results are illustrated in Figure 2. For WT-103, we observe trends similar to those of PTB; A large drop in perplexity corresponds to a much smaller decrease in R@3 (see Table 2, rows 3 and 5).

## 5.1 ACCURACY–EFFICIENCY TRADEOFFS

We illustrate the accuracy–efficiency tradeoff space of the PTB and WT-103 models in Figure 2. For each model, we tabulate the results at fixed intervals according to the approximated percentage of FLOPs, relative to that of the unpruned model. We omit results that exceed 100 in test perplexity, since they are insufficient for language modeling in practice.

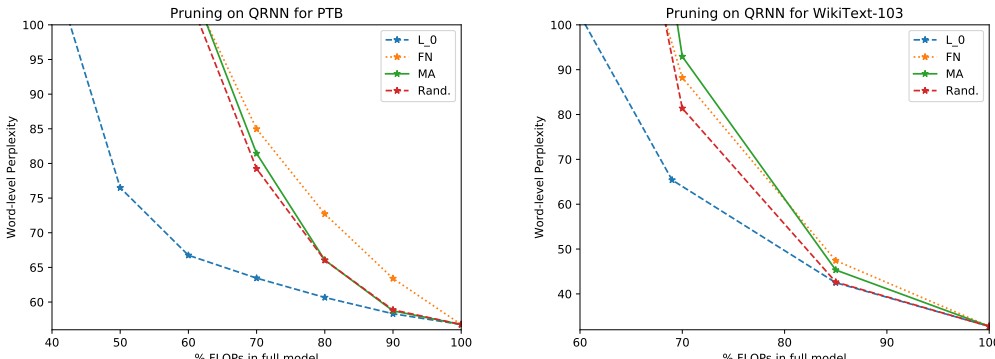

Figure 2: Full experimental results on Penn Treebank and WikiText-103. We illustrate the perplexity–efficiency tradeoff space on the test set obtained before applying the single-rank update.

Surprisingly, random filter pruning is a strong baseline, which supports the findings from Mittal et al. (2018). Random pruning not only outperforms filter norm and mean activation pruning, but also regains perplexity more easily with a single-rank update. From Table 1 (rows 7–9) and Table 2 (rows 6–8), we see that random pruning displays equivalent or superior performance to filter norm and mean activation pruning. Interestingly, random pruning achieves the lowest perplexity with a single-rank update (Table 2, rows 5–8), out of all the baseline approaches on WT-103.

On the other hand, filter norm pruning is relatively weak, doing worse than random pruning in all cases—with or without a single-rank update—suggesting that filter norm pruning has no practical benefit over random pruning. $L_0$ regularization (Louizos et al., 2018) works best, as shown in rows 5–6 in Table 1 and row 5 in Table 2.

In general, testing on Penn Treebank and WikiText-103—two very different datasets—gives us consistent results, thus demonstrating the robustness of $L_0$ regularization (Louizos et al., 2018), compared to the other pruning approaches.

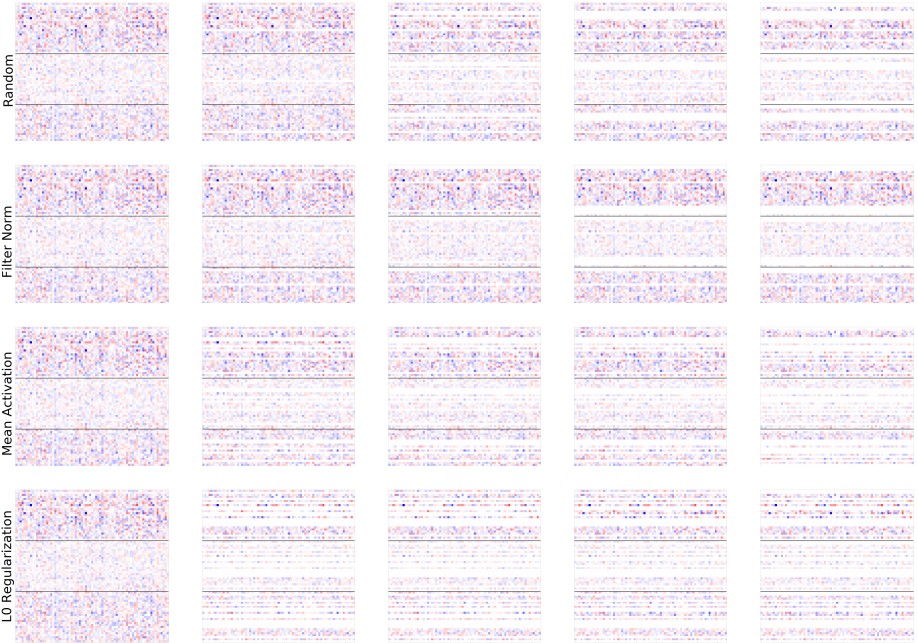

Figure 3: Illustration depicting pruning on a truncated subset of the first layer's weights from the PTB model, where each row corresponds to a different technique, and each column a different operating point. From left to right, the operating points are 100%, 80%, 70%, 60%, and 50% FLOPs. For each of the subfigures, we concatenate from top to bottom the first 25 filters of $\mathbf{W}_{\{z,f,o\}}$, and from left to right the first 75 elements in each filter, yielding square visualizations. All the pruning techniques appear to be dropping weights differently—we note that, for $L_0$ regularization (row 4), the dropped weights remain largely constant throughout.

## 5.2 POWER USAGE AND LATENCY

On the Raspberry Pi, the PTB models are relatively fast, while the WT-103 models are high latency, taking over one second (Table 2, rows 2–3 and 8) for the full models. For type-ahead prediction on a mobile device, the WT-103 models are unsuitable as-is; further steps (e.g., more pruning then re-training, vocabulary reduction, quantization) would be required to deploy the models for practical use. Supporting the findings from Tang et al. (2018), the number of FLOPs scales linearly with latency and power: Full experimental results from Figure 2 yield Pearson's $r^2 = 0.98$ for both latency– and power–FLOPs measurements, suggesting a strong linear relationship between the number of FLOPs and both latency and power.

In terms of extra parameters, a single-rank update costs less than 74 KB for `ptb-qrnn`, and less than 120 KB for `wt103-qrnn`. Mean activation statistics requires 20 KB for `ptb-qrnn`, and 30 KB for `wt103-qrnn`. Mask parameters for $L_0$ regularization cost about 20 KB on each power level for `ptb-qrnn`, and 30 KB for `wt103-qrnn`. Filter norm pruning and random pruning do not require any extra storage.

## 5.3 EFFECTS OF PRUNING ON INDIVIDUAL WORD-LEVEL RECALL

According to Zipf's law, a small set of common words disproportionately outnumbers a much larger set of rare words, suggesting that rare words require more capacity to model. Since pruning reduces model expressiveness by removing parameters, we hypothesize that rare words are more affected than common words are. To validate this hypothesis, we compare the recall of each individual word between our original and pruned models. Specifically, we collect the test set R@3 statistics of each present word in the vocabulary, order the words by rank, and then remove all words with zero R@3 in the original model. The last step is done to limit our attention to successfully modeled words, and

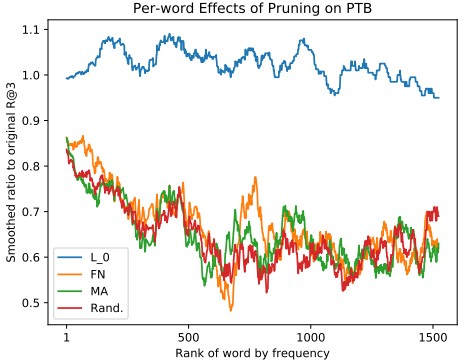 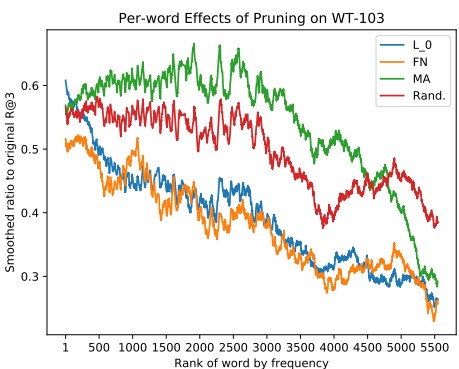

Figure 4: Visualization of the relative changes in per-word R@3 on the test set in the pruned models. We present results for 80% and 70% FLOPs pruning on the PTB and WT-103 models, respectively.

to avoid division-by-zero. Next, to determine the relative difference in R@3 between the models, we compute the per-word R@3 ratio between the pruned and original models. Finally, we apply valid convolution of a fixed-window, uniform averaging filter across the *x*-axis to construct a less noisy visualization.

As shown in Figure 4, filter norm and random pruning yield models with worse R@3 on rare words than common words (see top 100 words on both datasets). Mean activation pruning is similar to filter norm and random pruning on PTB; however, on WT-103, words with frequency ranks roughly between 500 and 3000 are penalized less than the most common 200 words are. The best method, $L_0$ regularization, surprisingly improves R@3 for many words on PTB. On WT-103, however, it yields curves similar to those from filter norm pruning. Overall, the graphs support the hypothesis that rare words are more difficult to model.

### 5.4 STUDY LIMITATIONS

We use the official PyTorch QRNN implementation,[5] which provides a fused pooling kernel for CUDA-enabled GPUs but lacks one for CPUs; such a kernel would provide further performance gains on a CPU. Since we perform our evaluation with a non-fused kernel, our latency results could be improved further in practice. Nevertheless, we successfully capture the general trend in accuracy–efficiency tradeoffs for NLMs, and we demonstrate the impact of inference-time pruning methods.

### 6 CONCLUSION AND FUTURE WORK

Motivated by the mass adoption of smart software keyboards on mobile devices, we explore the task of inference-time pruning on QRNNs, state-of-the-art neural language models. Starting with existing training-time pruning methods, we extend their usability to QRNNs at run-time, obtaining multiple operating points in the accuracy–efficiency tradeoff space. To recover some perplexity using a negligible amount of memory, we propose to train and store single-rank weight updates at desired operating points.

In this paper, we limit our inference-time power usage and latency evaluation to the RPi, a convenient surrogate for smartphones. A potential extension is to conduct a similar study on NLMs running on GPUs—in practice, this has applications in large-scale vocabulary speech recognition (Cho & Kumar, 2018). Clearly, the GPU architecture differs greatly from the RPi, favoring high concurrency and minimal host–device data transfer. Other extensions include evaluating other compression techniques on NLMs, such as knowledge distillation (Hinton et al., 2015) and vector quantization (Han et al., 2016).

---

[5] https://github.com/salesforce/pytorch-qrnn/

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
