# OpenReview forum: "Adaptive Pruning of Neural Language Models for Mobile Devices"
_ICLR.cc/2019/Conference_

### Official Review · AnonReviewer2 · 2018-10-31
**Torn About This Work**

**Rating:** 6
**Confidence:** 4

**Review:**

In this paper, the authors investigate the accuracy-efficiency tradeoff for neural language models. In particular, they explore how different compression strategies impact the accuracy (and flops), and more interestingly, also how it impacts the power use for a RaspberryPi. The authors consider the QRNNs and SRUs for this purpose and use standard datasets for their analysis. I am torn about this paper. On one hand, I feel that the analysis is interesting, thoughtful and detailed; the power usage statistics bring a different perspective to the compression community. The section on inference time pruning was especially interesting to read. On the other hand however, there is limited novelty in the setup. The authors use standard, well known, compression algorithms on common neural language modeling architectures and datasets and use out-of-the-box tools for their ultimate analysis. Further, the paper needs additional work before it can be accepted in my opinion. I detail my arguments below:

- The authors begin by discussing SwiftKey and similar apps but I'm not sure if its clear that they use neural language modeling as the backend. Do the authors have a source to validate this claim?
- Knowledge distillation is another algorithm that has been found to be quite competitive in compressing models into smaller versions of themselves. Have the authors experimented with that?
- The R@3 is an good metric but I suggest that the authors look at mean reciprocal rank (MRR) instead. This removes the arbitrary-ness of "3" while ensuring that the metric of interest is the accuracy and not probability of being correct (perplexity).
- Can you comment on the sensitivity of the results to the RPi frameworks? For instance, the RPi deployment tools, architecture, and variance in the predictions?
- Along the same line, I'm curious how generalizable the RPi results are for other computing architectures. For those of us who are not experts on hardware, it would be nice to read about whether similar tradeoffs will exist in other architectures such as mobile phones, GPUs or CPUs.
- Could the authors add some meta-analysis about the results? If the perplexity goes up as a consequence of compression, what kinds of tokens it that localized to? Is it primarily rare words that the model is less confident about, or are the probabilities for most words getting skewed?
- Finally, I feel that such an exploration will catch on only if the tools are open-sourced and made easy to replicate/use. If there were a blog or article summarizing the steps needed to replicate the power measurement (including sources from where to buy the necessary hardware), more people would be inclined on adding such an analysis to future neural language modeling work.

I am willing to revisit my rating, as necessary, once I read through the rebuttal.


UPDATE:

After reading the rebuttal, I am increasing my score to 6. The authors alleviated some of my concerns but my major concerns about their novelty and the impact of their results remains.

---

> ### Author Response · Authors · 2018-11-25
> **Re: Torn About This Work**
>
> We would like to thank the reviewer for their thorough feedback and interest in our work. We have taken the time to address your comments in the revision:
>
> - The authors begin by discussing SwiftKey and similar apps but I'm not sure if its clear that they use neural language modeling as the backend. Do the authors have a source to validate this claim?
>
> Yes, the official blog contains a post about using neural networks: https://blog.swiftkey.com/swiftkey-debuts-worlds-first-smartphone-keyboard-powered-by-neural-networks/. We have added this link as a footnote in the introduction.
>
> - Knowledge distillation is another algorithm that has been found to be quite competitive in compressing models into smaller versions of themselves. Have the authors experimented with that?
>
> Knowledge distillation is indeed interesting; it could be a potential extension of this work. We have mentioned it as future work in the conclusion.
>
> - The R@3 is an good metric but I suggest that the authors look at mean reciprocal rank (MRR) instead. This removes the arbitrary-ness of "3" while ensuring that the metric of interest is the accuracy and not probability of being correct (perplexity).
>
> Our justification for using R@3 is that most mobile keyboards provide three side-by-side predictions, since screen space is quite limited. We have added this reason to the revision.
>
> - Can you comment on the sensitivity of the results to the RPi frameworks? For instance, the RPi deployment tools, architecture, and variance in the predictions?
> - Along the same line, I'm curious how generalizable the RPi results are for other computing architectures. For those of us who are not experts on hardware, it would be nice to read about whether similar tradeoffs will exist in other architectures such as mobile phones, GPUs or CPUs.
>
> It would be interesting to extend this work to GPUs, whose operating characteristics are quite different from those of CPUs. For example, host-device data transfer and parallelism are much bigger issues on GPUs. An evaluation on GPUs is important due to the usage of NLMs in large-scale vocabulary speech recognition. We have added this in the conclusion and future work section of the revision.
>
> - Could the authors add some meta-analysis about the results? If the perplexity goes up as a consequence of compression, what kinds of tokens it that localized to? Is it primarily rare words that the model is less confident about, or are the probabilities for most words getting skewed?
>
> We have added a meta-analysis to the revision (see last subsection in the revised discussion). We find that the pruned model is less confident about rare words, and we hypothesize that the effects are due to common words being easier to model.
>
> - Finally, I feel that such an exploration will catch on only if the tools are open-sourced and made easy to replicate/use. If there were a blog or article summarizing the steps needed to replicate the power measurement (including sources from where to buy the necessary hardware), more people would be inclined on adding such an analysis to future neural language modeling work.
>
> We agree and plan to release our power measurement software upon the unblinding of this paper.
>
> We thank the reviewer again for the insightful comments and feedback.

---

### Official Review · AnonReviewer3 · 2018-11-02
**Nice work but I think another baseline is needed.**

**Rating:** 5
**Confidence:** 3

**Review:**

This paper proposes to evaluate the accuracy-efficiency trade off in QRNN language model though pruning the filters using four different methods. During evaluation, it uses energy consumption on a Raspberry Pi as an efficiency metric. Directly dropping filters make the accuracy of the models worse. Then the paper proposes single-rank update(SRU) method that uses negligible amount of parameters to recover some perplexity. I like this paper focuses on model's performance on real world machines.

1) The proposed approaches just work for QRNN, but not for many other neural language models such as LSTM, vanilla RNN language models, the title could be misleading.

2) In the experiment section, I think one baseline is needed for comparison: the QRNN language model with a smaller number of filters trained from scratch. With this baseline, we can see if the large number of filters are needed even before pruning.

---

> ### Author Response · Authors · 2018-11-25
> **Re: Nice work but I think another baseline is needed.**
>
> We thank the reviewer for the feedback and interest in our work. We would like to address your comments:
>
> 1) The proposed approaches just work for QRNN, but not for many other neural language models such as LSTM, vanilla RNN language models, the title could be misleading.
>
> We believe the title correctly reflects the work, since we perform adaptive pruning of NLMs on mobile devices. In addition to being efficient to train and deploy, QRNNs represent state of the art in NLM, with comprehensive analysis done in the literature (see https://arxiv.org/pdf/1803.08240.pdf).
>
> 2) In the experiment section, I think one baseline is needed for comparison: the QRNN language model with a smaller number of filters trained from scratch. With this baseline, we can see if the large number of filters are needed even before pruning.
>
> We agree; we have added the requested baselines to the results tables in the revision. We find that the smaller models indeed perform worse than the original models do.
>
> We thank the reviewer again for the insightful feedback.

---

### Official Review · AnonReviewer1 · 2018-11-05
**Much-needed exploration of efficiency tradeoffs in neural language model deployment**

**Rating:** 6
**Confidence:** 4

**Review:**

This paper presents an investigation of perplexity-efficiency tradeoffs in deploying a QRNN neural language model to mobile devices, exploring several kinds of weight pruning for memory and compute savings. While their primary effort to evaluate pruning options and compare points along the resulting tradeoff curves doesn't result in a model that would be small and fast enough to serve, the authors also introduce a clever method (single-rank weight updates) that recovers significant perplexity after pruning.

But there are many other things the authors could have tried that might have given significantly better results, or significantly improved the results they did get (the top-line 40% savings for 17% perplexity increase seems fairly weak to me). In particular:

- The QRNN architecture contains two components: convolutions alternate with a recurrent pooling operation. The fact that the authors report using a PyTorch QRNN implementation (which runs on the Arm architecture but doesn't contain a fused recurrent pooling kernel for any hardware other than NVIDIA GPUs) makes me afraid that they used a non-fused, op-by-op, approach for the pooling step, which would leave potentially 10 or 20 percentage points of free performance on the table. The QRNN architecture is designed for a situation where you already have optimized matrix multiply/convolution kernels, but where you're willing to write a simple kernel for the pooling step yourself; at the end of the day, pooling represents a tiny fraction of the QRNN's FLOPs and does not need to take more than 1 or 2 percent of total runtime on any hardware. (If you demonstrate that your implementation doesn't spend a significant amount of time on pooling, I'm happy to bump up my rating; I think this is a central point that's critical to motivating QRNN use and deployment).

- Once pooling is reduced to <2% of runtime, improvements in the convolution/matmul efficiency will have increased effect on overall performance. Perhaps your pruning mechanisms improved matmul efficiency by 50%, but the fact that you're spending more time on pooling than you need to has effectively reduced that to 40%.

- Although the engineering effort would be much higher, it's worth considering block-sparse weight matrices (as described in Narang et al. (Baidu) and Gray et al. (OpenAI)). While this remains an underexplored area, it's conceivable that block-sparse kernels (which should be efficient on Arm NEON with block sizes as low as 4x4 or so) and blockwise pruning could give more than a 50% speedup in convolution/matmul efficiency.

- In a real-world application, you would probably also want to explore quantization and distillation approaches to see if they have additional efficiency gains. Overall results of 10x or more wall clock time reduction with <5% loss in accuracy are typical for domains that have seen more optimization for mobile deployment (especially mobile-optimized CNNs like MobileNet), so I think that's entirely possible for your application.

---

> ### Author Response · Authors · 2018-11-26
> **Re: Much-needed exploration of efficiency tradeoffs in neural language model deployment**
>
> We thank the reviewer for the helpful comments:
>
> - The QRNN architecture contains two components: convolutions alternate with a recurrent pooling operation. The fact that the authors report using a PyTorch QRNN implementation (which runs on the Arm architecture but doesn't contain a fused recurrent pooling kernel for any hardware other than NVIDIA GPUs) makes me afraid that they used a non-fused, op-by-op, approach for the pooling step [...]
>
> We acknowledge that this is a limitation of our study -- a fused kernel would indeed provide performance gains. Nevertheless, we successfully capture the trends in the accuracy-efficiency tradeoffs of different inference-time pruning methods. To make this limitation aware to readers and practitioners, we have included this as part of a "study limitations" subsection in the revision (see last subsection in the discussion).
>
> - Although the engineering effort would be much higher, it's worth considering block-sparse weight matrices (as described in Narang et al. (Baidu) and Gray et al. (OpenAI)). While this remains an underexplored area, it's conceivable that block-sparse kernels (which should be efficient on Arm NEON with block sizes as low as 4x4 or so) and blockwise pruning could give more than a 50% speedup in convolution/matmul efficiency.
> - In a real-world application, you would probably also want to explore quantization and distillation approaches to see if they have additional efficiency gains. Overall results of 10x or more wall clock time reduction with <5% loss in accuracy are typical for domains that have seen more optimization for mobile deployment (especially mobile-optimized CNNs like MobileNet), so I think that's entirely possible for your application.
>
> Other compression approaches are quite interesting. We have mentioned vector quantization and distillation as potential future work in the revised conclusion.
>
> We thank the reviewer again for the insightful feedback.

---

### Public Comment · (anonymous) · 2018-11-08
**Recent related work**

Hey,  I agree nlm is powerful but sometimes too slow. You may be interested in a recent work also on language model pruning (as a contextualized representation model instead of a nlm).

Liyuan Liu, et, al.  "Efficient Contextualized Representation: Language Model Pruning for Sequence Labeling", EMNLP'18

Thanks

---

> ### Author Response · Authors · 2018-11-26
> **Re: Recent related work**
>
> Thanks for sharing this paper with us!

---

### Meta-Review · Area_Chair1 · 2018-12-02
**Reject**

**Confidence:** 4
**Recommendation:** Reject

**Metareview:**

The area chair agrees with the authors and the reviewers that the topic of this work is relevant and important. The area chair however shares the concerns of the reviewers about the setup and the empirical evaluation:
- Having one model that can be pruned to varying sizes at run-time is convenient, but in practice it is likely to be OK to do the pruning at training time. In light of this, the empirical results are not so impressive.
- Without quantization, distillation and fused ops, the value of the empirical results seems questionable as these are important and well-known techniques that are often used in practice. A more thorough evaluation that includes these techniques would make the paper much stronger.